# Ground state selection under pressure in the quantum pyrochlore magnet $Yb_2Ti_2O_7$

E. Kermarrec[1,2,3], J. Gaudet[2], K. Fritsch[4], R. Khasanov[5], Z. Guguchia[5], C. Ritter[6], K.A. Ross[7], H.A. Dabkowska[8] & B.D. Gaulin[2,8,9]

A quantum spin liquid is a state of matter characterized by quantum entanglement and the absence of any broken symmetry. In condensed matter, the frustrated rare-earth pyrochlore magnets $Ho_2Ti_2O_7$ and $Dy_2Ti_2O_7$, so-called spin ices, exhibit a classical spin liquid state with fractionalized thermal excitations (magnetic monopoles). Evidence for a quantum spin ice, in which the magnetic monopoles become long range entangled and an emergent quantum electrodynamics arises, seems within reach. The magnetic properties of the quantum spin ice candidate $Yb_2Ti_2O_7$ have eluded a global understanding and even the presence or absence of static magnetic order at low temperatures is controversial. Here we show that sensitivity to pressure is the missing key to the low temperature behaviour of $Yb_2Ti_2O_7$. By combining neutron diffraction and muon spin relaxation on a stoichiometric sample under pressure, we evidence a magnetic transition from a disordered, non-magnetic, ground state to a splayed ferromagnetic ground state.

[1] Laboratoire de Physique des Solides, CNRS, Univ. Paris-Sud, Université Paris-Saclay, Orsay Cedex 91405, France. [2] Department of Physics and Astronomy, McMaster University, Hamilton, Ontario, Canada L8S 4M1. [3] Laboratoire National des Champs Magnétiques Intenses, CNRS, Grenoble BP 166-38042, France. [4] Helmholtz-Zentrum Berlin für Materialien und Energie, Hahn-Meitner-Platz 1, Berlin 14109, Germany. [5] Laboratory for Muon Spin Spectroscopy, Paul Scherrer Institut, Villigen PSI CH-5232, Switzerland. [6] Institut Laue Langevin, BP 156, Grenoble 38042, France. [7] Department of Physics, Colorado State University, Fort Collins, Colorado 80523-1875, USA. [8] Brockhouse Institute for Materials Research, McMaster University, Hamilton, Ontario, Canada L8S 4M1. [9] Canadian Institute for Advanced Research, 180 Dundas St. W., Toronto, Ontario, Canada M5G 1Z8. Correspondence and requests for materials should be addressed to E.K. (email: edwin.kermarrec@u-psud.fr) or to B.D.G. (email: bruce.gaulin@gmail.com).

The pyrochlore lattice, comprised of corner-sharing tetrahedra, is the archetype of magnetic frustration in three dimensions[1] (Fig. 1). Since its early study by Anderson in 1956 (ref. 2), frustrated spin Hamiltonians on the pyrochlore lattice have provided a seemingly inexhaustible source for the study of fundamental physics[3,4]. In particular, spin liquid ground states have been predicted for such a lattice decorated with Heisenberg[5,6] or XXZ[7] spins. More recently, pyrochlore magnets have been put forward as realistic vehicles for the realization of a quantum spin ice state, using the generic $S = \frac{1}{2}$ nearest-neighbour anisotropic exchange Hamiltonian[8–10]. $Yb_2Ti_2O_7$ is a promising quantum spin ice candidate as it possesses both an (effective) $S = \frac{1}{2}$ spin, thanks to the well isolated crystal field Kramers doublet ground state appropriate to $Yb^{3+}$[11], and strong quantum fluctuations brought by anisotropic exchange interactions and an XY g-tensor[12]. Several studies have focused on the nature of the ground state in $Yb_2Ti_2O_7$, yet a consensus has been elusive to date[13–18]. Early neutron scattering experiments ruled out the presence of conventional static order down to 90 mK in a polycrystalline sample[15], whereas other single crystal studies concluded the ground state was ferromagnetic[14,16]. The results of local probes are even more puzzling. Muon spin relaxation (μSR) measurements evidenced the presence of true static moments on the muon timescale, through the observation of both a drop of asymmetry and a decoupling of the muon spins in longitudinal applied fields[18], along with a drastic slowing down of the fluctuation rate below $T_c$ for certain samples[13]. In contrast, μSR studies by D'Ortenzio et al.[17] found a non-magnetic, fluctuating ground state, in both stoichiometric polycrystalline and single crystal samples, despite the presence of pronounced specific heat anomalies at $T_c = 265$ mK and $T_c = 185$ mK, respectively. It is clear that local defects, either oxygen vacancies[19] or excess magnetic ions[20] (referred to as stuffing), vary significantly between polycrystalline powders and single crystals, and are likely responsible for such sample dependencies.

Here, by applying hydrostatic pressure to well-characterized $Yb_{2+x}Ti_{2-x}O_{7+\delta}$ samples, with $x = 0$ and $x = 0.046$ (ref. 20), we observe a magnetic transition in the stoichiometric, $x = 0$ sample from a disordered ground state into a splayed ferromagnetic ground state. This result sheds light on the origin of the sample dependence in the ground state selection for $Yb_2Ti_2O_7$ and is consistent with the recent theoretical proposal that $Yb_2Ti_2O_7$ lies close to a phase boundary in the generic quantum spin ice Hamiltonian phase diagram[21].

## Results

**Muon spin relaxation.** μSR measurements under hydrostatic pressures as high as 25 kbar, and at temperatures as low as 0.245 K, were performed on $Yb_{2+x}Ti_{2-x}O_{7+\delta}$ samples, with $x = 0$ and $x = 0.046$, at the GPD beamline of PSI. The muons are implanted inside the bulk of the material, and act as local magnetic probes. The signal coming from the muons that stop inside the pressure cell was measured separately and subtracted (see Supplementary Figs 1 and 2) from the overall signal.

Figure 2a shows the temperature dependence of the μSR signal for the stoichiometric, $x = 0$ sample in zero field, $R_{zf}(t)$, as a function of time $t$ and under an applied pressure $P = 19.7$ kbar. Well above $T_c = 0.265$ K, at $T \geq 0.97$ K, the majority of the $Yb^{3+}$ magnetic moments are paramagnetic and in a fast fluctuating regime, and display single-exponential relaxation. For $T \leq 0.5$ K, we observe the development of a small magnetic fraction $f$ of the $Yb^{3+}$ moments, which grows nonlinearly as the temperature decreases. The absence of oscillations at short time is indicative of a highly disordered magnetic state. The zero-field relaxation is well described by a Gaussian distribution of static internal fields with standard deviation $\Delta$ (see Supplementary Note 1) and the following phenomenological function:

$$R_{zf}(t) = f\left(\frac{2}{3}e^{-\Delta^2 t^2/2} + \frac{1}{3}e^{-\lambda t}\right) + (1-f)e^{-\lambda t} \qquad (1)$$

In a purely static scenario, the second term (1/3-tail) should be constant. Here, a fluctuating component is nonetheless observed and we modelled this using a relaxation rate $\lambda$. The third term accounts for the paramagnetic component, and assumes the same relaxation rate $\lambda$, for simplicity. The unconventional shape of the zero-field longitudinal relaxation was discussed in detail in refs 13,22. In contrast, the evolution of the relaxation in temperature of the $x = 0$ sample under zero applied pressure, shown in Fig. 2b, shows little or no magnetic fraction ($f \simeq 6\%$) at any temperature above our base $T = 0.245$ K, in agreement with D'Ortenzio et al.[17] previously reported μSR studies. Using equation (1) we extract the magnetic fraction for each pressure and temperature, and collect the results in Fig. 2c. The development of the magnetic fraction with temperature is clearly pressure dependent, and turns on strongly at low temperatures, below $T_c = 0.265$ K, for our minimum pressure of 1.2 kbar. For each pressure, one can define a critical temperature $T_c$, such that for $T \leq T_c$, 50% of the magnetic moments are frozen. The corresponding $P - T$ phase diagram is shown in Fig. 3. Clearly, the phase transition extrapolated from finite pressure measurements to zero pressure agrees well with the sharp $C_p$ anomaly at $T_c = 0.265$ K, appropriate to the $x = 0$ sample. However the zero-pressure state for the $x = 0$ sample at 0.245 K, below $T_c$, is disordered, indicating that the ground state of the stoichiometric, $x = 0$ sample, is a spin liquid.

We now turn to the $x = 0.046$ sample. The zero-field relaxation at $T = 0.245$ K under zero and an applied pressure $P = 24.1$ kbar are shown in Fig. 2d. Strikingly, no frozen magnetic fraction is observed upon the application of a pressure as high as $P = 24.1$ kbar. Instead, we observe an increase of the relaxation for this $x = 0.046$ sample, demonstrating its sensitivity to pressure. The temperature dependence of the relaxation is reported in Fig. 2e,f. One can speculate that a transition to a fully ordered state, as it is observed for the $x = 0$ sample, would require higher pressures or lower temperatures, consistent with the lower $T_c = 0.185$ K of the $x = 0.046$ sample.

μSR studies on other samples have reported a drastic slowing down of spin fluctuations[13], or static order[18], under zero applied pressure for temperatures below 0.25 K. In the light of our results, even relatively low (applied or chemical) pressure can destroy the

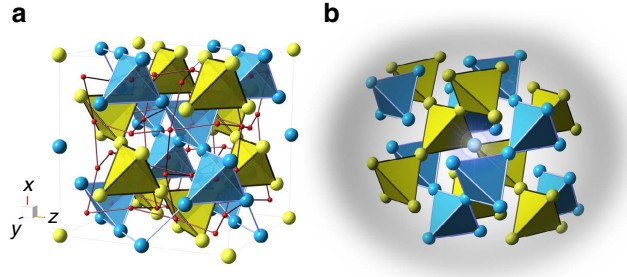

**Figure 1 | Pyrochlore structure of $Yb_{2+x}Ti_{2-x}O_{7+\delta}$.** Excess $Yb^{3+}$ ion can occupy a $Ti^{4+}$ site and create a local defect (referred to as 'stuffing'). (**a**) Representation of the ideal pyrochlore structure of $Yb_2Ti_2O_7$, with Yb in blue, Ti in yellow and O in red. Yb and Ti form corner-sharing tetrahedra lattices. (**b**) Schematic representation of the structurally distorted environment of a defect.

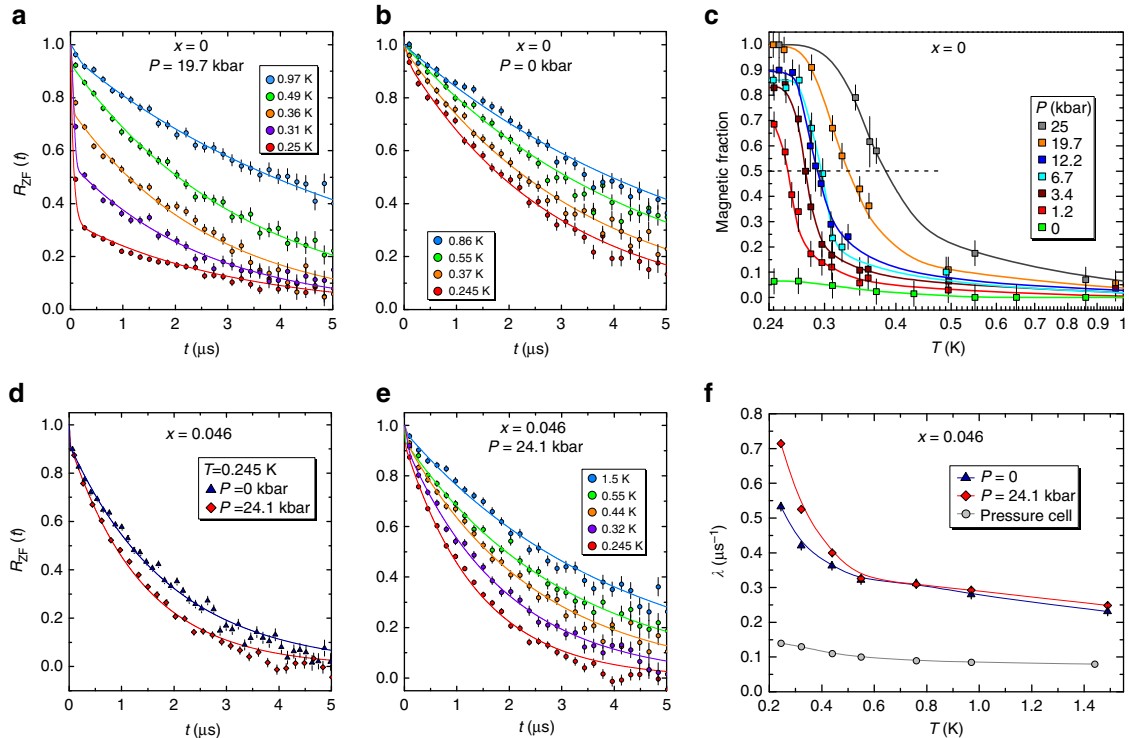

**Figure 2 | Temperature evolution of the μSR relaxation in Yb$_{2+x}$Ti$_{2-x}$O$_{7+\delta}$ under pressure.** (**a–c**) Refer to the $x = 0$ and (**d–f**) refer to the $x = 0.046$ sample. (**a**) The drastic increase of relaxation observed upon decreasing temperature in the $x = 0$ sample indicates a spin freezing under an applied pressure $P = 19.7$ kbar, which is absent under zero pressure (**b**) and for the $x = 0.046$ sample (**d**). (**c**) The temperature evolution of the magnetic fraction is reported for various pressures. The black horizontal dashed line represents a volume magnetic fraction of 50%, used as a criterion to define $T_c$. (**e,f**) For the $x = 0.046$ sample, only a moderate increase of the spin dynamics is observed under applied pressure. The error bars of the μSR relaxation data are of statistical origin and correspond to the square root of the total number of detected positrons resulting from muon decays. Error bars of the relaxation rate $\lambda$ represent standard deviation of the fit parameters. Error bars of the magnetic fraction represent standard deviation of the fit parameters, with a minimal value of 0.05 corresponding to the typical error on the total asymmetry for μSR under pressure.

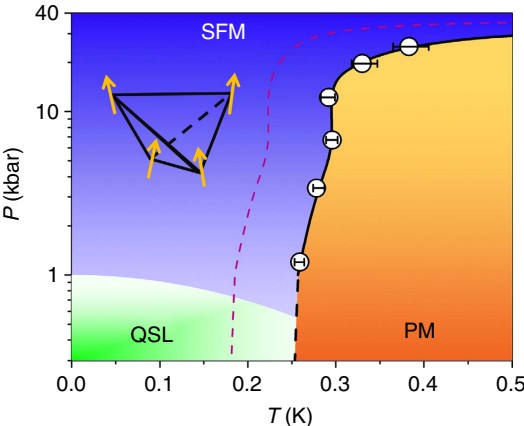

**Figure 3 | Pressure–temperature phase diagram of Yb$_{2+x}$Ti$_{2-x}$O$_{7+\delta}$.** The vertical axis displays the pressure $P$ in logarithmic scale and the horizontal axis the temperature $T$. Empty black circles define the transition line between the collective paramagnetic (PM, orange) and the splayed ferromagnetic (SFM, blue) regions relative to the $x = 0$ sample. The transition temperature is defined such that for $T \leq T_c$ 50% of the magnetic moments are frozen (see Fig. 2c). Error bars allow $T_c$ to be defined between 40 and 60% of the magnetic fraction. The green region highlights the presence of a disordered, non-magnetic phase (QSL) found at $P = 0$. Black thick line is a guide to the eye. Dashed purple line is the hypothetical transition line for $x = 0.046$.

disordered spin liquid state and induce magnetic order. A low level of defects in the different samples is a natural explanation to the contradictory μSR results. Such disorder, at the ~2% level, is difficult to characterize, but it is largely absent in polycrystalline samples, synthesized at lower temperatures by solid state methods.

**Neutron diffraction.** Armed with the knowledge of the $P − T$ phase diagram in Fig. 3, we sought to determine the nature of the pressure-induced magnetic order in Yb$_{2+x}$Ti$_{2-x}$O$_{7+\delta}$ samples, with $x = 0$, by performing neutron diffraction on the stoichiometric powder sample at the D20 high-flux diffractometer of the ILL. The detection of small magnetic moments under pressure using neutron diffraction is challenging due to the significant background signal of the pressure cell itself. Figure 4a shows the neutron diffraction data for the maximum hydrostatic pressure of the cell, $P = 11(2)$ kbar, and temperatures from 400 to 100 mK, from which a background measured at 800 mK was subtracted. We clearly observe the development of magnetic Bragg intensities at the (111), (311), (222) and (004) positions upon cooling below 400 mK. This is firm evidence for the existence of long-range magnetic order in Yb$_{2+x}$Ti$_{2-x}$O$_{7+\delta}$ samples, with $x = 0$, under an applied pressure $P = 11(2)$ kbar. The refinement of the neutron diffraction data gives us the temperature dependence of the ordered moment, shown in Fig. 4b. The contrast with previous experiments under zero pressure is striking. First, the saturated moment $\mu = 0.33(5)$ $\mu_B$ is

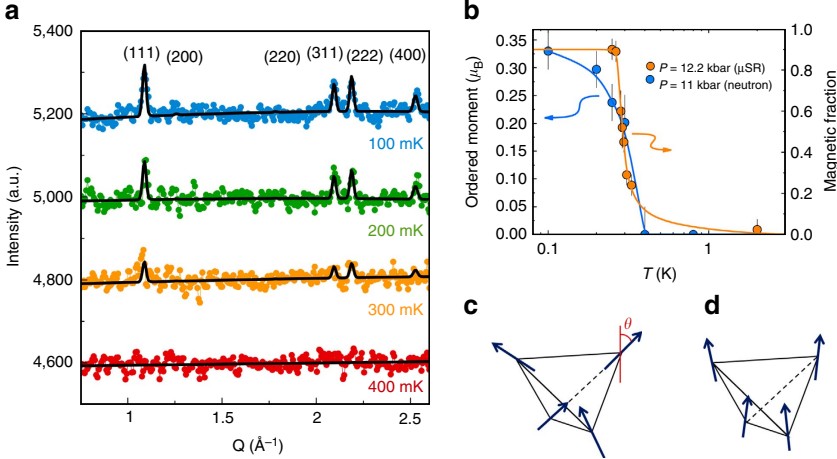

**Figure 4 | Neutron diffraction measurements of Yb₂Ti₂O₇ under applied pressure.** ($a$) Diffraction data sets from 400 to 100 mK after the 800 mK data set has been subtracted. Error bars are not shown for clarity. ($b$) Ordered moment versus temperature determined by neutron diffraction for $P = 11(2)$ kbar (blue, left axis) and magnetic fraction determined by μSR for $P = 12.2$ kbar (orange, right axis). Error bars of the ordered moment represent s.d. of the refinement. Schematic spin structure of the ice-like splayed ferromagnet with $\theta = 14°$ for $P = 0$ ($c$) and $\theta = 5°$ for $P = 11$ kbar ($d$), where $\theta$ is the splay angle between the [001] direction and the magnetic moment, tilted towards the local [111] directions of the tetrahedron.

much smaller than that $\mu \sim 1\ \mu_B$ reported previously for different Yb₂Ti₂O₇ samples[16], although similar to the ordered moment in the $\Gamma_5$ ordered state of Yb₂Ge₂O₇ (ref. 23). Second, the ordered moment vanishes cleanly above $T_c \sim 0.4$ K, with no anomalous magnetic Bragg intensity well above $T_c$ (refs 24,25). The previously reported order parameter at $P = 0$ of our $x = 0$ polycrystalline sample is anomalous[24]; it shows no change across $T_c$ and only falls off at much higher temperatures. Consistency with our $P = 0$ μSR results on the same sample requires that this Bragg-like scattering is dynamic on slow time scales. That notwithstanding, the magnetic structure previously refined on the basis of a very high temperature ($\sim 8$ K) background subtraction gave a splayed ice-like ferromagnetic structure[24], with the moments on a tetrahedron lying mainly in the [100] direction with a positive splay angle $\theta = 14(5)°$, such that the moments tilt towards the local [111] direction (Fig. 4c). The components perpendicular to the local [111] axis obey the 2-in/2-out ice rule on a single tetrahedron. A different type of splayed ferromagnet, with the perpendicular components satisfying the all-in/all-out structure, has also been reported recently[26], in addition to a nearly collinear ferromagnet ($\theta \sim 0°$)[16], for other samples. The magnetic structure associated with the true Bragg scattering we refine here in the stoichiometric $x = 0$ sample under $P = 11(2)$ kbar is also a splayed ice-like structure, but with a much reduced splay angle $\theta = 5(4)°$, such that it is close to a collinear [100] ferromagnet (Fig. 4d).

## Discussion

These results bring a fresh perspective on the long standing debate about the presence or absence of static magnetic order in the quantum pyrochlore magnet Yb₂₊ₓTi₂₋ₓO₇₊δ. The acute sensitivity to local (through the Yb³⁺ stuffing) or applied pressure is surprising. However, a corollary of our new $P - T$ phase diagram is that non-stoichiometric samples with non-zero chemical pressure can easily display an ambient applied pressure phase transition to a splayed ferromagnetic state at $T_c$. Yet, this interpretation remains challenged by the fact that our $x = 0.046$ sample does not show evidence for magnetic order at ambient pressure, and by previous reported observations of a magnetic transition in polycrystalline, likely $x \sim 0$, sample even under zero pressure[13,18]. This may indicate that the non-magnetic low-temperature region of the phase diagram is extremely

narrow, existing only for a certain range of $x$, whose absolute values are still to be determined. This would actually be reminiscent of the recent findings on the Tb₂₊ₓTi₂₋ₓO₇₊δ pyrochlore magnet, which has been shown to display an ordered phase that is extremely sensitive to disorder, appearing only for $0 < x < 0.01$ (refs 27–29).

Furthermore, the present work illustrates the relevance of applying hydrostatic pressure to tune the magnetic properties of frustrated pyrochlore compounds, a path that was followed by pioneering work on the other spin liquid candidate Tb₂Ti₂O₇ (ref. 30). In case of Yb₂Ti₂O₇, we found that the pressure tunes the delicate balance between the anisotropic exchanges of the quantum spin ice Hamiltonian, and selects a splayed ferromagnetic ground state away from the degenerate antiferromagnetic ground states manifold. This scenario confirms recent theoretical proposals that Yb₂Ti₂O₇ lies close to phase boundaries derived from the generic $S_{eff} = \frac{1}{2}$ quantum spin ice Hamiltonian[21], and provides the missing key to understand its exotic magnetic properties. Particularly appealing is the prediction that accidental degeneracies in the vicinity of these phase boundaries can lead to the emergence of a quantum spin liquid[31]. This would offer a natural explanation for a non-magnetic, disordered state under zero pressure in stoichiometric Yb₂Ti₂O₇ and recent observations of a continuum of gapless quantum excitations[24,32] at low temperatures.

## Methods

**Sample preparation.** The Yb₂₊ₓTi₂₋ₓO₇₊δ samples with $x = 0$ and $x = 0.046$ were prepared at the Brockhouse Institute for Materials Research, McMaster University. The $x = 0$ powder sample was obtained through conventional solid-state reaction between pressed powders of Yb₂O₃ and TiO₂ sintered at 1,200 °C in air. The $x = 0.046$ powder sample was obtained by crushing a single crystal grown by the floating zone method in 4 atm of O₂ with a growth rate of 5 mm h⁻¹. More details on the details of the synthesis and the characterization can be found in ref. 25.

**Muon spin relaxation.** μSR measurements were carried out at the GPD instrument of the Paul Scherrer Institut, Switzerland. About 1 g of each powder sample was mixed with $\sim 3$ mm³ of a pressure medium (Daphne 7373 oil) and placed inside the sample channel of a double-wall pressure cell. Two different cells were used, labelled as (1) and (2) (see Supplementary Note 1), and are described in more details in ref. 33. The muon momentum was adjusted in order to obtain an optimal fraction of the muons stopping in the sample, with optimal values found at $P = 106$ and $P = 107$ MeV c⁻¹. The relaxation of both cells were measured without any sample down to 0.245 K. The applied pressure was determined by measuring

the superconducting transition temperature of a small piece of pure indium inserted in the sample channel[33].

**Neutron diffraction.** The neutron diffraction experiment was conducted at the D20 beamline, a high intensity two axis diffractometer, at the Institut Laue-Langevin, using a neutron wavelength $\lambda = 2.4$ Å. A mass of 1.5 g of $Yb_2Ti_2O_7$ powder sample and a small amount of NaCl powder, which serves as a pressure calibration, were both mounted in a high pressure clamp cell and inserted in a $^3$He–$^4$He dilution fridge. Fluorinert was used as a pressure transmitter. A minimum of 12 h of data was collected at each temperature. The diffraction pattern obtained for $T = 800$ mK is shown in Supplementary Fig. 3. Structural refinements for both NaCl and $Yb_2Ti_2O_7$ and magnetic refinements for $Yb_2Ti_2O_7$ have been performed using the Fullprof program suite[34].

**Data availability.** The data sets generated during and/or analysed during the current study are available from the corresponding author on reasonable request.

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

## Acknowledgements

Work at McMaster University was supported by NSERC of Canada. This work is based on experiments performed at SμS, Paul Scherrer Institute, Villigen, Switzerland and at the Institut Laue-Langevin, Grenoble, France. This project has received funding from the European Union's Seventh Framework Programme for research, technological development and demonstration under the NMI3-II Grant number 283883. E.K. acknowledges useful discussions with P. Mendels, F. Bert, S. Petit, L. D. C. Jaubert and C. Decorse.

## Authors contributions

E.K., J.G. and B.D.G. wrote the manuscript. E.K., J.G., K.F. and B.D.G. performed the neutron diffraction experiment. E.K. and B.D.G. performed the μSR experiment. K.A.R. and H.A.D. synthesized and characterized the samples. C.R. designed and performed the neutron scattering experiment. Z.G. and R.K. designed and performed the μSR experiment. All the co-authors discussed the results and improved the manuscript.

## Additional information

**Competing interests:** The authors declare no competing financial interests.

**Publisher's note**: 

