## [Peer review file · Nature Communications]

Reviewers' comments:

Reviewer #1 (Remarks to the Author):

This manuscript address the question of ground-state order in the three-dimensional quantum spin liquid candidate, $\text{Yb}_2\text{Ti}_2\text{O}_7$, and identifies a transition, under pressure, from a disordered to a ferromagnetically ordered ground state. The fact that some samples order under ambient pressure is argued to be a consequence of non-stoichiometry, resolving a long-standing dispute about the intrinsic nature of the ground state of $\text{Yb}_2\text{Ti}_2\text{O}_7$.

The quest to find and characterise quantum spin liquids has become an important theme in condensed matter physics. The majority of materials discussed as three-dimensional spin-liquids are pyrochlore oxides, and among these, $\text{Yb}_2\text{Ti}_2\text{O}_7$ shows particular promise as a material with a small magnetic moment and large quantum fluctuations. However, despite being intensively studied, $\text{Yb}_2\text{Ti}_2\text{O}_7$ remains an enigma.

All neutron scattering studies carried out on $\text{Yb}_2\text{Ti}_2\text{O}_7$ to date shows evidence of correlated fluctuations of Yb moments, in the form of highly-structured diffuse scattering at low temperatures, and propagating spin excitations in high magnetic field. In some samples, these correlated fluctuations persist to the lowest temperatures measured, suggesting a spin-liquid ground state. However other samples, especially those prepared as a powder, show clear thermodynamic evidence of a phase transition for $T \sim 250$ mK. Moreover, a number of studies have presented evidence of a "splayed" ferromagnetic ground state, which has been argued to be arise as a result of a Higgs transition out of a high-temperature spin liquid. Crudely put, while all samples of $\text{Yb}_2\text{Ti}_2\text{O}_7$ do something interesting, it is not clear that all samples do the same interesting thing. And this mysterious sample dependence remains a serious obstacle to understanding possible quantum spin-ice behaviour in $\text{Yb}_2\text{Ti}_2\text{O}_7$.

In this manuscript, Kermarrec et al. argue that the intrinsic ground state of $\text{Yb}_2\text{Ti}_2\text{O}_7$ is a quantum spin liquid, and that the key to understanding why some samples show ferromagnetic order lies in the pressure-dependence of their properties. In support of this claim, Kermarrec et al. present the results of μSR and neutron scattering studies of powder samples of $\text{Yb}_{2+x}\text{Ti}_{2-x}\text{O}_{7+\delta}$ with two different stoichiometries, $x=0$ and $x=0.046$, previously studied by Ross et al., in Ref. 19.

Evidence for an ordered moment is taken from μSR experiments, while structural refinement of elastic neutron scattering experiments confirms the expected "splayed ferromagnetic" order. Their main discovery is that samples of very high chemical purity, which do not order at ambient pressure, can be driven to order ferromagnetically by the application of pressure. The authors note interesting parallels with recent theoretical work emphasising proximity to a phase boundary as the reason for the unusual properties of $\text{Yb}_2\text{Ti}_2\text{O}_7$.

There is no question that this manuscript contains a significant new study of an important material. If the interpretation offered by Kermarrec et al. is correct, it will lay to rest a long-

standing problem in understanding the sample-dependence of different studies on $\text{Yb}_2\text{Ti}_2\text{O}_7$. Resolving this issue would leave us one step closer to understanding a three-dimensional quantum spin liquid, an question of greater significance than the material itself. And, regardless of interpretation, the possibility of tuning between ordered and disordered states with pressure is an exciting discovery, which could make a big impact on the field. For of all these reasons, I believe that these results are worthy of publication in Nature Communications.

Given that the nature of the ground state in $\text{Yb}_2\text{Ti}_2\text{O}_7$ has already proved controversial, these results are likely to prompt a lively debate, and I imagine that not all will immediately be won over by Kermarrec et al.'s arguments. One point where I am personally curious is the correlation between the effect of pressure and disorder on the sample. In an earlier paper [Ref. 19] the authors state :

"The lattice spacing of the crushed crystal sample is larger than the sintered powder (see Fig. 5). This is generally consistent with stuffing in pyrochlores [where] the increased cell size simply arises to accommodate the larger rare-earth ions in the titanium sites."

I am not sure how to reconcile this with the present observation that high-purity samples, under pressure, act like samples of lower purity at ambient pressure. None the less, I do not think that this amounts to a critical objection, since the effect of stuffing in real materials may extend far beyond lattice spacing. And the experimental possibility of switching between a spin liquid and a ferromagnet, as a function of pressure, remains exciting in any case.

In conclusion, I have no hesitation in recommending this manuscript for publication in Nature Communications. The experimental evidence for magnetic order in some samples of $\text{Yb}_2\text{Ti}_2\text{O}_7$ is convincing, and its pressure-dependence is a significant new finding. I believe that the interests of those working on spin liquids are best served by the prompt publication of this manuscript in a high-profile journal, where it can stimulate a widespread discussion. The manuscript is clearly written, and I do not believe that it needs any substantial revision before publication. But for completeness, I include a list of minor points which the referees might like to consider when revising the manuscript.

1. (With apologies for seeming pedantic). In the abstract the authors refer to a quantum spin-liquid as a state with long-range entanglement. While long-range entanglement is certainly a reason to be excited about quantum spin ice, there are many different forms of quantum spin liquid, including some gapped spin liquids with only short-range entanglement.

A famous example of such a spin liquid, where entanglement is restricted to nearest-neighbour bonds with a 4-site unit cell, is given in Shastry and Sutherland, *Physica B+C* 108, 1069 (1981).

2. In line 21 the authors refer to the spin liquid in a pyrochlore lattice "decorated with Heisenberg spins", and cite the work of Hermele et al., Ref. 13.

I think this statement also risks causing confusion : the well-known results of Hermele et al.

for a $U(1)$ quantum spin liquid were derived for an XXZ model on the pyrochlore lattice.

The Heisenberg model on a pyrochlore lattice is also believed to support a spin liquid ground state. However, for classical spins, the relevant gauge group is $U(1) \times U(1) \times U(1)$ - see e.g.

C.L. Henley

Phys. Rev. B 71, 014424 (2005). Establishing the gauge group in the case of quantum spins remains an open problem.

3. (Not so important). There appear to be minor inconsistencies in the way in which hyphens are used in different parts of the text (cf. "zero pressure state", line 74 and "zero-field relaxation", line 76).

4. The authors include a brief discussion, beginning at line 109, of a recent article by Yaouanc et al. [Ref. 24] which claims to observe a different form of FM order. To the best of my knowledge, the refinement of order described in this manuscript is consistent - a "splayed FM" with a net moment along the [100] axis, and a small, symmetric, canting of the spins into the [100] plane - is consistent with all existing theories of $\text{Yb}_2\text{Ti}_2\text{O}_7$. I am uncertain how order of the type described by Yaouanc et al. could arise in a system where single-ion anisotropy is forbidden.

5. In line 128, the authors refer to " $U(1)$ manifold phases". I cannot find where this terminology is defined in the text. Has it perhaps been borrowed from Jaubert et al. [Ref. 10] ?

Reviewer #2 (Remarks to the Author):

The authors present a combined muon spin relaxation and neutron diffraction study of the quantum spin ice candidate material $\text{Yb}_2\text{Ti}_2\text{O}_7$. Specifically, they show a pressure-induced transition to a magnetically ordered state in a stoichiometric sample that is not observed in a sample with a small amount of excess Yb, and argue this sensitivity to applied and chemical pressure explains the strong sample dependence of the observed properties. This is an important result, as it addresses the puzzling problem of sample dependence and suggests an avenue through which to tune the ground state properties of this interesting material.

However, I am not convinced the authors have fully justified their rather strong conclusion. A number of the specific claims made in support of the conclusion are unclear or inconsistent. For these reasons, I cannot recommend publication in the current form. Specific comments follow.

1) I struggle with the claim that pressure dependence provides the sole explanation for the observed sample dependence. I ask that the authors consider softening their claim or that they discuss other possible explanations and why those explanations have been discounted. For example, in reference [22] from the same group, it was postulated that sample dependence could be explained by a difference in magnetic anisotropy of the Yb^{3+} ions on the Ti site.

2) The μSR data show a pressure-induced transition in the $x = 0$ sample that is absent in the $x = 0.046$ sample. The authors argue that the phase boundary to the ordered state is moved to higher pressures and lower temperatures in the disordered sample. However, they also argue that this could explain the observation of an ordered state under zero applied pressure in a different sample. This would require the phase boundary to move in the opposite direction of that suggested by the previous argument.

3) On p. 5, the authors claim disorder is "largely absent in polycrystalline samples, synthesized at lower temperatures by solid state methods," but the contradictory μSR studies cited in the same paragraph were both performed on polycrystalline samples synthesized by solid state methods.

4) The comparison of the high pressure neutron diffraction data presented in this work to the zero pressure neutron diffraction data presented previously (Ref 22) leaves out a number of important details.

a. First, a distinction is drawn between the fitted saturated moments. However, the $P = 0$ measurement used 8 K data as background, while the $P = 11$ kbar measurement used 800 mK data as background. Most of the increase observed at the Bragg locations at $P = 0$ happened above 700 mK. What would the fitted saturated moment be if the 700 mK data were used as background instead?

b. Second, the authors claim no anomalous magnetic intensity is observed above T_c in the $P = 11$ kbar measurement. Do the authors expect to be able to resolve the anomalous intensity if it is there, given the extra background signal due to the pressure cell, and given that the highest measured temperature was 800 mK?

5) The distinction being drawn between the magnetic structures at $P = 0$ and $P = 11$ kbar is

unclear. Are these not the same representation with slightly different splay angles? Figure 4 c) and d) along with the Fig. 4 caption make it seem like they are being described as distinct structures.

Reviewers' comments:

We thank Reviewer #1 for his/her thorough comments and suggestions for our manuscript.

Reviewer #1 (Remarks to the Author):

This manuscript address the question of ground-state order in the three-dimensional quantum spin liquid candidate, Yb₂Ti₂O₇, and identifies a transition, under pressure, from a disordered to a ferromagnetically ordered ground state. The fact that some samples order under ambient pressure is argued to be a consequence of non-stoichiometry, resolving a long-standing dispute about the intrinsic nature of the ground state of Yb₂Ti₂O₇.

The quest to find and characterise quantum spin liquids has become an important theme in condensed matter physics. The majority of materials discussed as three-dimensional spin-liquids are pyrochlore oxides, and among these, Yb₂Ti₂O₇ shows particular promise as a material with a small magnetic moment and large quantum fluctuations. However, despite being intensively studied, Yb₂Ti₂O₇ remains an enigma.

All neutron scattering studies carried out on Yb₂Ti₂O₇ to date shows evidence of correlated fluctuations of Yb moments, in the form of highly-structured diffuse scattering at low temperatures, and propagating spin excitations in high magnetic field. In some samples, these correlated fluctuations persist to the lowest temperatures measured, suggesting a spin-liquid ground state. However other samples, especially those prepared as a powder, show clear thermodynamic evidence of a phase transition for $T \sim 250$ mK. Moreover, a number of studies have presented evidence of a "splayed" ferromagnetic ground state, which has been argued to be arise as a result of a Higgs transition out of a high-temperature spin liquid. Crudely put, while all samples of Yb₂Ti₂O₇ do something interesting, it is not clear that all samples do the same interesting thing. And this mysterious sample dependence remains a serious obstacle to understanding possible quantum spin-ice behaviour in Yb₂Ti₂O₇.

In this manuscript, Kermarrec et al. argue that the intrinsic ground state of Yb₂Ti₂O₇ is a quantum spin liquid, and that the key to understanding why some samples show ferromagnetic order lies in the pressure-dependence of their properties. In support of this claim,

Kermarrec et al. present the results of muSR and neutron scattering studies of powder samples of $\text{Yb}_{2+x}\text{Ti}_{2-x}\text{O}_{7+\delta}$ with two different stoichiometries, $x=0$ and $x=0.046$, previously studied by Ross et al., in Ref. 19.

Evidence for an ordered moment is taken from muSR experiments, while structural refinement of elastic neutron scattering experiments confirms the expected "splayed ferromagnetic" order. Their main discovery is that samples of very high chemical purity, which do not order at ambient pressure, can be driven to order ferromagnetically by the application of pressure. The authors note interesting parallels with recent theoretical work emphasising proximity to a phase boundary as the reason for the unusual properties of $\text{Yb}_2\text{Ti}_2\text{O}_7$.

There is no question that this manuscript contains a significant new study of an important material. If the interpretation offered by Kermarrec et al. is correct, it will lay to rest a long-standing problem in understanding the sample-dependence of different studies on $\text{Yb}_2\text{Ti}_2\text{O}_7$. Resolving this issue would leave us one step closer to understanding a three-dimensional quantum spin liquid, a question of greater significance than the material itself. And, regardless of interpretation, the possibility of tuning between ordered and disordered states with pressure is an exciting discovery, which could make a big impact on the field. For all these reasons, I believe that these results are worthy of publication in Nature Communications.

Given that the nature of the ground state in $\text{Yb}_2\text{Ti}_2\text{O}_7$ has already proved controversial, these results are likely to prompt a lively debate, and I imagine that not all will immediately be won over by Kermarrec et al.'s arguments. One point where I am personally curious is the correlation between the effect of pressure and disorder on the sample. In an earlier paper [Ref. 19] the authors state :

"The lattice spacing of the crushed crystal sample is larger than the sintered powder (see Fig. 5). This is generally consistent with stuffing in pyrochlores [where] the increased cell size simply arises to accommodate the larger rare-earth ions in the titanium sites."

I am not sure how to reconcile this with the present observation that high-purity samples, under pressure, act like samples of lower purity at ambient pressure. None the less, I do not think that this amounts to a critical objection, since the effect of stuffing in real

materials may extend far beyond lattice spacing. And the experimental possibility of switching between a spin liquid and a ferromagnet, as a function of pressure, remains exciting in any case.

If one assumes crudely that the replacement of Ti by a larger Yb only shortens the surrounding Yb—Yb chemical bonds in a homogeneous way, then such a stuffing can actually be seen as an equivalent of a positive, hydrostatic, pressure. If this is true, one can imagine that a stuffed sample with non-zero x should have some internal strain that could have the same effect than hydrostatic pressure and thus could induce a magnetic ordering, even at “ambient” pressure. This is one possible interpretation that can be drawn from our results.

Yet, this interpretation seems to be inconsistent with the fact that our $x=0.046$ sample does not show a conventional magnetic ordering at ambient pressure, or with the reports of other groups that observed a magnetic transition in a pure, $x=0$, sample even under zero pressure.

In our opinion there are two possibilities that can explain such inconsistency.

1) The non-magnetic region is extremely narrow within the phase diagram, existing only for a certain range of x values. The uncertainty of the composition of a nominally “ $x=0$ ” sample could be still too important to allow meaningful comparison between samples of different origin. This would actually be reminiscent of the recent findings on the $Tb_{2+x}Ti_{2-x}O_{7+\delta}$ pyrochlore magnet, that has been shown to display a non-dipolar ordered phase that is extremely sensitive to disorder, appearing only for $0 < x < 0.01$ [PRB **92**, 245114 and PRB **87**, 060408(R)].

2) The effect of the substitution of Ti by Yb cannot be understood as an increase of the lattice parameter/positive pressure only, and the exact local distortion—including, but not limited to, its impact on the local crystal field—should be considered as well.

While our present experimental results clearly demonstrate the sensitivity of the ground state of $Yb_2Ti_2O_7$ to applied pressure, we believe that a systematic thorough study of a set of stuffed samples $Yb_{2+x}Ti_{2-x}O_{7+\delta}$ with different x would be required to solve this issue.

In conclusion, I have no hesitation in recommending this manuscript for publication in Nature Communications. The experimental evidence for magnetic order in some samples of $Yb_2Ti_2O_7$ is convincing, and its pressure-dependence is a significant new finding. I believe that the interests of those working on spin liquids are best served by the prompt publication of this manuscript in a high-profile journal, where it can stimulate a widespread discussion. The manuscript is clearly written, and I do not believe that it needs any substantial revision before publication. But for completeness, I include a list of minor points which the referees might like to consider when revising the manuscript.

1. (With apologies for seeming pedantic). In the abstract the authors refer to a quantum spin-liquid as a state with long-range entanglement. While long-range entanglement is certainly a reason to be excited about quantum spin ice, there are many different forms of quantum spin liquid, including some gapped spin liquids with only short-range entanglement.

A famous example of such a spin liquid, where entanglement is restricted to nearest-neighbour bonds with a 4-site unit cell, is given in Shastry and Sutherland, *Physica B+C* 108, 1069 (1981).

We thank the referee for the correction, and we have modified the sentence in the abstract accordingly: “A quantum spin liquid is a novel state of matter characterized by long **or short** range quantum entanglement and...”

2. In line 21 the authors refer to the spin liquid in a pyrochlore lattice "decorated with Heisenberg spins", and cite the work of Hermele et al., Ref. 13.

I think this statement also risks causing confusion : the well-known results of Hermele et al. for a $U(1)$ quantum spin liquid were derived for an XXZ model on the pyrochlore lattice.

The Heisenberg model on a pyrochlore lattice is also believed to support a spin liquid ground state. However, for classical spins, the relevant gauge group is $U(1)\times U(1)\times U(1)$ - see e.g. C.L. Henley *Phys. Rev. B* 71, 014424 (2005). Establishing the gauge group in the case of quantum spins remains an open problem.

In order to avoid confusion we now write:

“In particular, spin liquid ground states have been predicted for such a lattice decorated with Heisenberg\cite{Moessner1998,Canals1998} or XXZ\cite{Hermele2004} spins.”

The Heisenberg and the XXZ model are now clearly separated and assigned with the following references:

R. Moessner and J. T. Chalker, *Phys. Rev. Lett.* 80, 2929 (1998).

B. Canals and C. Lacroix, *Phys. Rev. Lett.* 80, 2933 (1998).

M. Hermele, M. P. A. Fisher, and L. Balents, *Phys. Rev. B* 69, 064404 (2004).

3. (Not so important). There appear to be minor inconsistencies in

the way in which hyphens are used in different parts of the text (cf. "zero pressure state", line 74 and "zero-field relaxation", line 76).

Thanks for pointing these inconsistencies. We have added a hyphen when the whole term is used as a compound adjective.

4. The authors include a brief discussion, beginning at line 109, of a recent article by Yaouanc et al. [Ref. 24] which claims to observe a different form of FM order. To the best of my knowledge, the refinement of order described in this manuscript is consistent - a "splayed FM" with a net moment along the [100] axis, and a small, symmetric, canting of the spins into the [100] plane - is consistent with all existing theories of Yb₂Ti₂O₇. I am uncertain how order of the type described by Yaouanc et al. could arise in a system where single-ion anisotropy is forbidden.

Indeed, Yaouanc et al. assumed that the crystal symmetry changes below T_c, going from the cubic Fd3m space group to the tetragonal Imma space group. The reported structural studies does not support such symmetry breaking at the moment. Nevertheless, the neutron diffraction pattern that they measured at 60mK shows some additional intensity at the 220 position as compared to the 1.5K data that is not present in our data. We thus considered that it is worth mentioning this work, given the sample dependence history of Yb₂Ti₂O₇.

5. In line 128, the authors refer to "U(1) manifold phases". I cannot find where this terminology is defined in the text. Has it perhaps been borrowed from Jaubert et al. [Ref. 10] ?

This terminology is indeed borrowed from Jaubert et al. and refers to the degenerate classical ground states of the anisotropic nearest-neighbors Hamiltonian on the pyrochlore lattice. The symmetry of the ground-state degeneracy is U(1) since it is generated by a continuous rotation of all spins. We have replaced this undefined term with the following sentence:

"...and selects a splayed ferromagnetic ground state away from **the degenerate antiferromagnetic ground states manifold**"

Reviewer #2 (Remarks to the Author):

We thank Reviewer #2 for his/her careful reading of our manuscript.

The authors present a combined muon spin relaxation and neutron diffraction study of the quantum spin ice candidate material Yb₂Ti₂O₇. Specifically, they show a pressure-induced transition to a magnetically

ordered state in a stoichiometric sample that is not observed in a sample with a small amount of excess Yb, and argue this sensitivity to applied and chemical pressure explains the strong sample dependence of the observed properties. This is an important result, as it addresses the puzzling problem of sample dependence and suggests an avenue through which to tune the ground state properties of this interesting material.

However, I am not convinced the authors have fully justified their rather strong conclusion. A number of the specific claims made in support of the conclusion are unclear or inconsistent. For these reasons, I cannot recommend publication in the current form. Specific comments follow.

1) I struggle with the claim that pressure dependence provides the sole explanation for the observed sample dependence. I ask that the authors consider softening their claim or that they discuss other possible explanations and why those explanations have been discounted. For example, in reference [22] from the same group, it was postulated that sample dependence could be explained by a difference in magnetic anisotropy of the Yb³⁺ ions on the Ti site.

In our manuscript, we put forward that our experimental finding on the pressure sensitivity of Yb₂Ti₂O₇ is the missing key to understand its sample-to-sample dependence. We underlined that it is an important finding, that is based on the observation of a pressure-induced magnetic transition for the x=0 sample, and on the absence of such transition for the x=0.046 sample (in the accessible temperature and pressure ranges). The fact that the two samples with different x behave differently under pressure directly evidences the impact of stuffing on the magnetic properties of Yb₂Ti₂O₇.

However we agree that the question of the microscopic origin of the impact of pressure is still open, and this is not addressed by our current work. We have changed the wording through the manuscript to better reflect this fact (changes are highlighted in blue).

Regarding the specific comment on the magnetic anisotropy of Yb³⁺, we believe reviewer #2 had the following reference in mind: PRB 92, 134420 rather than PRB 93, 064406. In this paper, we did not claim that the sample dependence could be explained by a difference in magnetic anisotropy of the Yb³⁺ ion. We instead showed that the magnetic anisotropy of **isolated** defects (whether it is Yb at the Ti site or due to oxygen vacancy) is Ising-like, and thus very different from the XY character of the majority of the Yb³⁺ ions. There is however a hint of the large “volume of influence” of such a local defect in the anomalously large energy width of the crystal field levels at the regular Yb³⁺ position, and that was ascribed to the existence of a relatively large strain field due to the defect, in agreement with our new results.

2) The muSR data show a pressure-induced transition in the x = 0 sample that is absent in the x = 0.046 sample. The authors argue that the phase boundary to the ordered state is moved to higher pressures and lower temperatures in the disordered sample. However, they also argue that this could explain the

observation of an ordered state under zero applied pressure in a different sample. This would require the phase boundary to move in the opposite direction of that suggested by the previous argument.

This issue is discussed before in our response to the comment of reviewer #1.

Since we did not observe any magnetic transition under pressure for the $x=0.046$ sample, we could only speculate that such transition would exist at lower temperature and/or for higher pressure and accordingly our text describes this possibility as a speculation only. However it is true that such possibility would seem inconsistent with the idea that samples from other groups showing an ordered state would be highly stuffed. As explained above, we speculate that this may be due to the limited extent of the spin-liquid region in the phase diagram and/or to the real effect of Yb stuffing that is more complicated than just applying chemical pressure.

Following on this comment made by our reviewers, we have now added a short paragraph to discuss this possibility at the end of the manuscript.

3) On p. 5, the authors claim disorder is "largely absent in polycrystalline samples, synthesized at lower temperatures by solid state methods," but the contradictory μ SR studies cited in the same paragraph were both performed on polycrystalline samples synthesized by solid state methods.

We note that the preparations of the samples are not identical. Chang et al. and Hodges et al. both reported a higher sintering temperature of respectively 1300°C and 1400°C, as compared to 1200°C in our case (incorrectly indicated as 1000°C in our earlier version). The sharpness of the lambda-type anomaly in the specific heat for the corresponding polycrystalline samples data is also different (see Fig.1 in Chang et al., PRB 89, 184416). The temperature is likely not the only factor that could influence the level of disorder and being able to precisely control the level of substitution in this, and many others, compounds remains a great challenge for solid state chemists and physicists.

We also like to emphasize that the level of defect we are dealing with, i.e. of the order of 2%, is extremely small. For this reason, comparing studies on different samples synthesized in even slightly different conditions is very difficult. In this regard, a reasonable approach is to study and compare samples that were synthesized under exactly the same conditions, after the same characterization technique has been applied to all of them.

4) The comparison of the high pressure neutron diffraction data presented in this work to the zero pressure neutron diffraction data presented previously (Ref 22) leaves out a number of important details.

a. First, a distinction is drawn between the fitted saturated moments. However, the $P = 0$ measurement used 8 K data as background, while the $P = 11$ kbar measurement used 800 mK data as background. Most of the increase observed at the Bragg locations at $P = 0$ happened above 700 mK. What would the fitted saturated moment be if the 700 mK data were used as background instead?

Reviewer #2 is right that the refined magnetic moment for both $P = 0$ and $P = 11$ kbar measurements cannot be directly compared. However, it is clear that the temperature dependence of the extracted moment for our previous $P=0$ and new $P=11$ kbar measurements are very different. For the $P=11$ kbar measurement the refined static moment is 0.33 μB whether the 800mK **or** the 400mK dataset is used as a background. In the case of our former $P=0$ measurement, the **gradual** increase of elastic intensity between 700mK and 100mK would correspond to roughly 0.15 μB (see the intensities reported in PRB **93**, 064406) and to almost zero (~ 0.04 μB) if one considers the increase of intensity between 100 and 400mK. The situation is thus clearly different.

b. Second, the authors claim no anomalous magnetic intensity is observed above T_c in the $P = 11$ kbar measurement. Do the authors expect to be able to resolve the anomalous intensity if it is there, given the extra background signal due to the pressure cell, and given that the highest measured temperature was 800 mK?

Again, for $P = 11$ k bar no intensity change has been observed between 400mK and 800mK and this is clearly different from our previous results at $P = 0$ for which an increase in the Bragg scattering intensity between these two temperature is observed (Fig.3 of PRB **93**, 064406). A refinement of the difference in intensity of the two diffraction patterns (400mK-800mK) give essentially zero within 0.05 μB of error bar (see additional Figure). We also recall that the transition temperature observed at $P=11$ kbar with neutron diffraction is in very good agreement with the μSR results, and that also clearly contrasts with the zero pressure case.

5) The distinction being drawn between the magnetic structures at $P = 0$ and $P = 11$ kbar is unclear. Are these not the same representation with slightly different splay angles? Figure 4 c) and d) along with the Fig. 4 caption make it seem like they are being described as distinct structures.

This is indeed the same magnetic order with only a different splay angle. For Fig.4d, the splay angle is 5 degree and thus very close to a pure ferromagnetic state for which the splay angle would be exactly zero. We have modified the caption to make it more obvious.

REVIEWERS' COMMENTS:

Reviewer #1 (Remarks to the Author):

NCOMMS-16-14398-T

"Ground-state selection under pressure in the quantum pyrochlore magnet $\text{Yb}_2\text{Ti}_2\text{O}_7$ "

E. Kermarrec et al.

In the first round of review, both Reviewers expressed a very positive opinion of the experimental findings in this manuscript.

In their first report, Reviewer #1 queried the relationship between physical pressure and chemical pressure arising from non-stoichiometry ("stuffing"), and raised a number of other minor points, but was content to recommend publication in Nat. Commun. on the strength of the experimental results.

Reviewer #2 also expressed puzzlement about the relationship between physical and chemical pressure. The Reviewer asked that the authors clarify this and a number of other points of interpretation before the manuscript could be accepted for publication.

In their Rebuttal, the authors address all of the points raised in the first round of review, and describe the corresponding revisions which they have made to the manuscript.

With regard to what is perhaps the most serious question, the authors acknowledge that the relationship between the effects of pressure and non-stoichiometry ("stuffing") is a complex one. The revised manuscript offers a more nuanced discussion of this point, acknowledging some of the challenges to the interpretation which the authors wish to present.

I am satisfied with these revisions, and those made in the light of the other question raised in the first report of Reviewer #1.

In addition, Reviewer #2 also raised questions about the microscopic consequences of "stuffing", sample synthesis, interpretation of neutron scattering data, and the nature of the ordered state.

Questions about the microscopic effects of chemical impurities go right to the heart of what we don't understand about $\text{Yb}_2\text{Ti}_2\text{O}_7$, and probably cannot be answered fully without a lot of further experimental and theoretical effort.

However I am satisfied that the authors have adequately answered the questions in the first report of Reviewer #2, within the limits of what is currently known about $\text{Yb}_2\text{Ti}_2\text{O}_7$.

In summary, the revised manuscript takes into account the criticisms made in the first round of review, and offers a slightly more balanced account of the authors findings. I am satisfied with the manuscript in this form, and remain of the opinion that these findings merit publication in Nat. Commun.

Reviewer #2 (Remarks to the Author):

The authors have addressed my major concerns. I can now recommend publication in Nature Communications.